# Transforaminal Endoscopic Lumbar Foraminotomy for Juxta-Fusional Foraminal Stenosis

**DOI:** 10.3390/jcm12175745

**Published:** 2023-09-04

**Authors:** Yong Ahn, Han-Byeol Park

**Affiliations:** Department of Neurosurgery, Gil Medical Center, Gachon University College of Medicine, Incheon 21565, Republic of Korea; phbsgood@gilhospital.com

**Keywords:** adjacent segment disease, endoscopic, foraminal stenosis, foraminotomy, fusion, lumbar, percutaneous, transforaminal

## Abstract

Adjacent segment foraminal stenosis is a significant adverse event of lumbar fusion. Conventional revision surgery with an extended fusion segment may result in considerable surgical morbidity owing to extensive tissue injury. Transforaminal endoscopic lumbar foraminotomy (TELF) is a minimally invasive surgical approach for symptomatic foraminal stenosis. This study aimed to demonstrate the surgical technique and clinical outcomes of TELF for the treatment of juxta-fusional foraminal stenosis. Full-scale foraminal decompression was performed via a transforaminal endoscopic approach under local anesthesia. A total of 22 consecutive patients who had undergone TELF were evaluated. The included patients had unilateral foraminal stenosis at the juxta-fusional level of the previous fusion surgery, intractable lumbar radicular pain despite at least six months of non-operative treatment, and verified pain focus by imaging and selective nerve root block. The visual analog scale and Oswestry Disability Index scores significantly improved after the two-year follow-up period. The modified MacNab criteria were excellent in six patients (27.27%), good in 12 (55.55%), fair in two (9.09%), and poor in two (9.09%), with a 90.91% symptomatic improvement rate. No significant surgical complications were observed. The minimally invasive TELF is effective for juxta-fusional foraminal stenosis.

## 1. Introduction

Lumbar fusion surgery is considered the standard procedure for degenerative spondylolisthesis or combined foraminal and intracanal stenosis. This procedure may provide sufficient decompression and immediate segmental stabilization, which can be maintained in the long term. However, it may also cause significant adjacent segment disease (ASD), defined as the subsequent development of spinal stenosis or instability at the juxta-fusional level [1,2,3,4,5,6,7,8]. Orita et al. [4] reported that, during an average follow-up of 13.3 months after floating fusion surgery, adjacent L5-S1 foraminal stenosis was found in eight of 125 patients (6.4%). Adjacent level foraminal stenosis is one of the primary causes of intractable radiculopathy after fusion surgery [9,10,11].

Surgical treatment options for juxta-fusional foraminal stenosis can be summarized into two categories: (1) extension of fusion surgery, and (2) open or minimally invasive foraminotomy. The first option is based on wide decompression with an extension of the fusion, which may simultaneously provide thorough decompression and stabilization. However, the invasiveness of this approach can lead to considerable morbidity or surgical trauma [12,13,14,15]. The second category is an open paraspinal approach with foraminotomy; nonetheless, this approach may cause serious facet injuries and lead to segmental instability progression [16,17,18]. For those reasons, a minimally invasive spine surgery (MISS) option may be required for these complicated cases.

Transforaminal endoscopic lumbar foraminotomy (TELF) is an efficient and minimally invasive surgical option for various foraminal stenosis [19,20,21,22,23]. We have also demonstrated our techniques and clinical results of TELF for primary lumbar foraminal stenosis [20,23,24]. This percutaneous procedure decompresses the nerve root while preserving the facet joint and segmental stability under local anesthesia. Recently, some authors have reported the use of the TELF technique for foraminal stenosis after fusion surgery [25,26,27]. The clinical outcome was favorable, and the authors emphasized the clinical usefulness of TELF in ASD. However, previously published studies on TELF are limited to case reports or endoscopic discectomy series for foraminal disc herniations. There is a paucity of longitudinal studies including sufficient cases and spanning extended follow-up periods.

In this study, we demonstrate the technique and clinical outcomes of TELF for adjacent-level foraminal stenosis after fusion surgery. Furthermore, we discuss the clinical importance of MISS approaches for ASD.

## 2. Materials and Methods

### 2.1. Patient Population

Between January 2018 and April 2021, 22 consecutive patients with intractable radiculopathy secondary stenosis underwent TELF. Patients were prospectively registered in our surgical database, and their medical records were retrospectively reviewed. The study was conducted in accordance with the Declaration of Helsinki and approved by the Institutional Review Board (GDIRB 2022-118), and written informed consent was obtained. The inclusion criteria were: (1) unbearable radicular leg pain despite more than three months of non-operative therapies; (2) at least 12 months of a pain-free period after the primary lumbar fusion surgery; (3) exiting nerve root (ENR) compression resulting from severe foraminal stenosis [28,29] at the level adjacent to the previous fusion surgery location, as demonstrated by computed tomography (CT) and magnetic resonance imaging (MRI); (4) stable foraminal stenosis without significant segmental instability on dynamic lateral X-rays; and (5) foraminal stenosis, as the source of the radiculopathy, which was proven by thorough neurologic examination and selective nerve root block to the ENR. Patients with isolated low back pain, acute disc herniation, severe central stenosis, segmental instability, or other pathological conditions, such as inflammation, infection, trauma or tumors, were excluded.

### 2.2. Surgical Procedure

The surgical procedure was primarily based on a previously demonstrated TELF method [23]. Full-scale foraminal decompression was achieved in three stages: (1) a fluoroscopy-guided transforaminal approach, (2) bony unroofing using endoscopic drill burrs, and (3) soft-tissue decompression using micro-punches (Figure 1). Preoperatively, intramuscular midazolam (0.05 mg/kg) and intravenous fentanyl (0.8 μg/kg) were administered on call. The patient was placed on a spinal table in the prone position with the hips and knees flexed.

#### 2.2.1. Fluoroscopy-Guided Transforaminal Approach

The fluoroscopy guidance aimed to ensure the safe placement of the working sheath at the foraminal zone. The outside-in approach was preferred over the inside-out method to prevent ENR irritation or injury during the procedure. The tip of the working sheath was located immediately in front of the ENR, and did not pass through the nerve root. The skin entry using the approach angle was determined based on preoperative imaging studies. Usually, the primary approach angle is approximately 45° or higher for full-scale foraminal decompression and endoscopic flexibility. However, the rod or screw head of the previous instrument could be in the approach route. In the patients included here, the approach trajectory was changed to a shallow angle. The location of the nerve root in the neural foramen and the severity of neural compression should be checked in the imaging studies. The initial trajectory should be directed to the foraminal pathologies while avoiding direct injury to the nerve root. First, an 18-gauge needle was inserted posterolateral to the foraminal zone. The needle tip targeted the disc surface or foraminal vertebral body, touching the surface of the superior articular process (SAP). After a preemptive epidural block through the needle, the guidewire was replaced, followed by serial obturator insertion, until the final obturator was placed at the foramen without any sign of irritation. The bevel-ended working sheath was fitted into the foramen with its sharp end away from the ENR using gentle mallet tapping (Figure 2A). Ideally, the working sheath should be firmly engaged in the foramen while avoiding the ENR to obtain a sufficient surgical field for foraminal decompression. Then, the working channel endoscope can be introduced to view the foraminal anatomies, including ENR with some perineural fat, disc surface, hypertrophic foraminal ligaments, and facet joints.

#### 2.2.2. Bony Unroofing under Endoscopic Visualization

This step focused on resecting the SAP and shoulder osteophytes to enlarge the bony foraminal aperture. The early endoscopic view included the ENR covered with perineural fat and the disc surface. The surface of the SAP was then exposed by rotating the endoscope and working sheath. The terminal part of the SAP was gradually resected using special endoscopic burrs and micro-punches until the ligamentum flavum (LF) and foraminal ligaments were adequately exposed. Proper exposure of the facet joint is essential for effective bony unroofing. Both SAP and the pedicle, including the synovium, should be exposed. Then, the SAP resection can proceed along the synovium surface. Finally, the surgeon can reach the axillary epidural space and LF. Bleedings from the resected bone surface and epidural veins were controlled using a radiofrequency coagulator tip and hemostatic agents (Figure 2B).

#### 2.2.3. Soft Tissue Decompression under Endoscopic Visualization

After sufficient foraminal unroofing and widening, the soft tissues, including the LF, foraminal ligaments, and redundant discs, were removed to release the pinched nerve root. When the tip of the SAP was sufficiently cleared to expose the ligamentous structures, delicate soft tissue decompression was performed. The hypertrophic LF, foraminal ligaments, or extruded discs were removed using forceps, micro-punches, Kerrison punches, and radiofrequency designed for endoscopic use. Most importantly, the main decompression was directed proximally and obliquely to the ENR route instead of parallel to the disc space. As the removal of soft tissue proceeded, ENR began to appear and was gradually released. Tissue adhesions were dissected using endoscopic probes and radiofrequency ablation. The removal and dissection procedures were conducted until the axillary epidural point at which the ENR started from the dural sac was exposed. Once the proximal axillary zone was identified, nerve root decompression was performed in the lateral exit zone. The remaining soft tissues or disc fragments were trimmed during full-scale foraminal decompression (Figure 2C).

#### 2.2.4. Endpoint and Postoperative Care

The TELF endpoint was determined by identifying the axillary epidural space and providing sufficient release to the ENR from the proximal to the lateral exit zone. Decompressed ENR was confirmed by solid pulsation based on the patient’s heartbeat (Figure 2D). Postoperatively, the surgeon assessed the patients’ general status for at least three hours to see if there were any neuromuscular symptoms or signs. The patients were discharged within 24 h in the absence of adverse events. Postoperative MRI or CT scans were used for precise pathological assessment, as required (Figure 3).

### 2.3. Outcome Evaluation

Data comprising a two-year follow-up period were collected during regular outpatient office visits and telephone surveys. Surgical outcomes were assessed using the visual analog scale (VAS) and Oswestry disability index (ODI) [30]. The global results were evaluated using the modified MacNab criteria [31]: excellent (free of pain, no restriction of activity), good (occasional non-radicular pain, presenting symptom relief), fair (improved functional capacity, but still handicapped), or poor (insufficient improvement, further operative intervention required). The recorded perioperative data included operative time, length of hospital stay, and complications.

### 2.4. Statistical Analysis

Statistical analyses were conducted by an independent statistician using SPSS (version 14.0; SPSS Inc., Chicago, IL, USA). Pre- and postoperative clinical data were compared using repeated-measures analysis of variance and paired *t*-tests. Statistical significance was set at *p* < 0.05.

## 3. Results

The mean age of the 22 patients included (12 female and 10 male) was 70.4 years (range, 54–91 years; Table 1). The mean duration between primary fusion surgery and the onset of radiculopathy was 50.32 months (range, 15–74 months). The mean symptom duration was 12.64 months (range, 3–25 months). The spinal levels intervened were L2-3 in one (4.5%) patient, L3-4 in seven (31.8%), L4-5 in eight (36.4%), and L5-S1 in six (27.3%). Thirteen (59.1%) patients had foraminal stenosis in the upper adjacent region, whereas the remaining nine (40.9%) had foraminal stenosis at the lower level adjacent to the previous fusion segment. The mean duration of operation was 60.9 min (range, 35–100 min). The mean postoperative hospital stay was 1.8 days (range, 1–5 days).

The VAS score (mean ± SD) for the lumbar radiculopathy significantly improved from 8.41 ± 0.85, preoperatively, to 3.36 ± 1.99, 2.95 ± 1.86, 2.09 ± 1.37, and 2.23 ± 1.34 at 6 weeks, 6 months, 1 year, and 2 years postoperatively, respectively (*p* < 0.001 Figure 4A). Additionally, the ODI score (mean ± SD) improved from 67.59 ± 17.17%, preoperatively, to 30.11 ± 16.58%, 28.65 ± 16.32%, 20.59 ± 15.30%, and 22.03 ± 16.62% at 6 weeks, 6 months, 1 year, and 2 years postoperatively, respectively (*p* < 0.001; Figure 4B).

The overall clinical outcomes according to the modified MacNab criteria were excellent in six patients (27.27%), good in 12 (54.55%), fair in two (9.09%), and poor in two (9.09%). Therefore, the rate of symptomatic improvement was 90.91% (Figure 5). Two of the 22 patients (9.09%) with poor outcomes experienced sustained radicular pain and postoperative flare-ups. These patients were managed with repeated postoperative blocks and oral medications. No significant postoperative complications were observed. Furthermore, no changes in segmental stability were documented in radiological studies during the follow-up period.

## 4. Discussion

### 4.1. Data Interpretation

Our results revealed that clinical improvement was relevant in pain scores, disability, and global functional status. The mean VAS score for radicular pain improved by 6.14 (74.01%) at the final follow-up (*p* < 0.001). The mean ODI improved by 45.57 (67.41%) at the final follow-up (*p* < 0.001). According to previous studies, a VAS score change > 50% [32] and an ODI reduction > 30% [33,34] are clinically significant. Thus, our TELF technique resulted in relevant outcomes for adjacent-segment foraminal stenosis following fusion surgery. The pain score and ODI steadily decreased until 1 year postoperatively and then slightly increased or stabilized. This phenomenon may be related to the natural course of degenerative changes over time.

### 4.2. Conventional Surgery for Adjacent Level Foraminal Stenosis

The conventional gold standard surgical option for adjacent-level foraminal stenosis is open decompression with additional fusion surgery. Another traditional surgical option is decompression surgery alone using an open paraspinal approach. Regardless of the surgery type, revision surgery may be complicated by fibrotic adhesions, incidental durotomy, neural injury, hematoma, or surgical site infection. Moreover, prolonged operative time under general anesthesia may cause various surgical complications and medical comorbidities [35]. Generally, the use of general anesthesia may increase the risk of adverse events, such as cardiovascular disorders, respiratory problems, blood loss, nausea, vomiting, and nerve damage [36,37,38]. Therefore, a less invasive and reliable alternative surgical technique is required for patients with adjacent segment problems.

### 4.3. TELF Technique for Juxta-Fusional Foraminal Stenosis

Based on the pathogenesis, ASD can be categorized into three disease entities: ASD due to central stenosis, ASD due to foraminal stenosis, and ASD due to herniated disc [11]. Theoretically, percutaneous endoscopic or key-hole decompression may be more suitable for foraminal stenosis than wide central stenosis [23]. Therefore, our endoscopic procedure focused on the foraminal pathologies at the juxta-fusional segment.

TELF has been developed as an effective and minimally invasive surgical option for lumbar foraminal stenosis. Knight et al. [39] introduced an endoscopic laser foraminoplasty technique using a side-firing laser beam for foraminal sculpturing. Percutaneous endoscopic foraminal decompression using a bone trephine with forceps has also been reported [20,40]. The current TELF type can be effectively applied to various lumbar foraminal stenoses using endoscopic burrs, steerable forceps, and micro-punches. A definitive full-scale foraminal decompression effect can be achieved with TELF for a broad spectrum of lumbar stenoses, including dynamic or severe foraminal stenosis with a collapsed disc [23].

Juxta-fusional foraminal stenosis can also be treated using this percutaneous endoscopic procedure. Wu et al. [26] reported the outcomes of transforaminal endoscopic foraminal decompression after lumbar fusion surgery, concluding that the transforaminal endoscopic approach could effectively decompress the neural foramen without further spinal destabilization. However, the surgical procedures were mainly performed for missed pathologies at the fused level. Yamashita et al. [27] published a case report describing a successful TELF technique for adjacent foraminal stenosis after lumbar fusion.

### 4.4. Pros and Cons of TELF

The benefits of TELF are significant. First, the percutaneous transforaminal approach provides an adequate angle of approximation to the narrowed foramen, avoiding previous scarring while preserving the facet joint. Therefore, it allows for full-scale foraminal decompression without additional fusion. Second, the procedure can be performed under local anesthesia with intravenous sedation. Therefore, direct injury to the ENR can be avoided by monitoring patient feedback. Finally, patients can return to their normal lives earlier. However, despite its minimal invasiveness and effectiveness, the steep learning curve and technical difficulties associated with TELF are its critical limitations. Surgeons can only obtain relevant and reliable outcomes after achieving technical proficiency. Therefore, the clinical use of TELF should be carefully considered.

### 4.5. Technical Keys to Success

Although TELF is effective for foraminal decompression, some specialized technical points should be considered, since the endoscopic surgical field is relatively limited and unfamiliar to surgeons. One of the most critical technical tips is that decompression, including bone resection and soft tissue removal, should be directed obliquely to the axillary point along the ENR. The direction of TELF differs from that of a transforaminal endoscopic lumbar discectomy (TELD). Decompression parallel to the disc space, as performed in TELD, cannot sufficiently release the ENR. Incorrect direction is one of the main reasons for surgical failure. Moreover, the initial endoscopic landing should be as close as possible to the ENR while avoiding nerve root irritation. Based on our experience, we recommend aiming at the posterior vertebral body surface near the upper endplate of the disc. The sharp end of the working sheath should be directed away from the ENR, allowing the surgeon to observe the epidural fat of the ENR in the initial endoscopic visual field without causing nerve root irritation. Finally, the surgeon must confirm the axillary epidural space and free mobilization of the ENR to complete the procedure.

Exposure to ENR alone is insufficient for full-scale foraminal decompression. Though the ENR can be observed during the process, even at an early stage, surgeons must continue to decompress the exposed ENR until the neural tissue is released. Once released, the ENR begins to beat according to the arterial blood flow and epidural pressure.

### 4.6. Limitations of the Study

Our study had some limitations. First, the study was conducted retrospectively without an appropriate control group. Therefore, a considerable selection bias might have been involved in patient enrollment. Second, the reduced number of patients precludes drawing credible conclusions. Third, the two-year follow-up period may have been relatively short for tracing long-term spinal stability at the treatment level. Therefore, a long-term prospective cohort study or randomized trial with a larger number of patients is required to verify the effectiveness of TELF in this particular ASD pathology.

Finally, we could not evaluate the natural course or incidence of ASD after fusion surgery. Although foraminal stenosis after fusion surgery comprises a minority of the pathologies seen in ASD, we strictly included adjacent segment foraminal stenosis cases in our study. First, most of our patients underwent primary surgery at another institute. Only three of the 22 patients underwent fusion surgery at our hospital. Therefore, we could not obtain the clinical and radiological data at the time of the primary surgery in most patients. Second, the main purpose of this study was to evaluate the clinical outcome of the TELF technique for juxta-fusional foraminal stenosis cases. Central stenosis, intracanalicular disc herniation, or spondylolisthesis was not indicated for this procedure. Third, this study was retrospective, and most patients lacked detailed primary data. Therefore, we could not demonstrate the postoperative course or incidence of ASD after the primary surgery.

### 4.7. Future Perspective

With increasing average life expectancy, the number of patients with adjacent segment disease after primary lumbar spine surgery is predicted to increase. Therefore, the need for endoscopic procedures performed under local anesthesia is increasing. Our data revealed that the clinical outcomes of TELF in post-spinal surgery syndrome are relevant and reliable. Surgical approaches, devices, and optical technologies have advanced remarkably. The working channel endoscope with a bigger working space enables more potent instruments. Steerable or navigable forceps and punches can reach the corner-side pathologies. Various articulating burrs will make the bony resection safer and more efficient. A high-definition monitor system will make the surgeon perform the procedure more precisely. The spinal navigation system will be applied to the endoscopic spine surgery to increase the accuracy of the surgery and to decrease the radiation exposure during the surgery. Eventually, the spine society will accept this kind of novel endoscopic technique as a mainstream spine surgery protocol to address patients’ needs.

## 5. Conclusions

TELF can be effective against foraminal stenosis at the level adjacent to the previous lumbar fusion. If there is no definitive segmental instability, full-scale foraminal decompression can be achieved using a percutaneous transforaminal endoscopic approach under local anesthesia, without requiring extensive revision surgery.

## Figures and Tables

**Figure 1 jcm-12-05745-f001:**
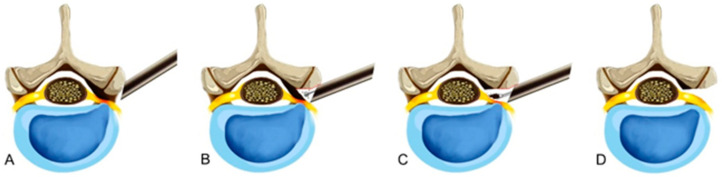
Schematic illustration of the transforaminal endoscopic lumbar foraminotomy procedure. (**A**) Foraminal placement of the working sheath using a percutaneous posterolateral approach. (**B**) Unroofing of the stenotic foramen by removing the superior articular process and osteophytes. (**C**) Release of the compressed nerve root by removing the ligamentum flavum and soft tissues. (**D**) Endpoint of the full-endoscopic foraminal decompression.

**Figure 2 jcm-12-05745-f002:**
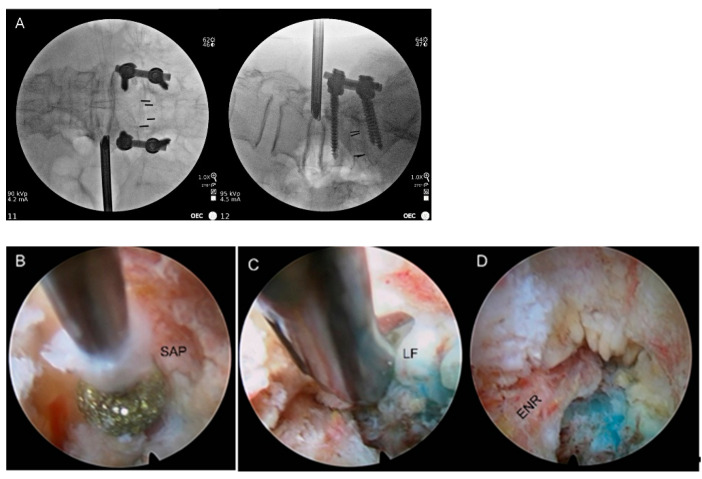
Intraoperative pictures of transforaminal endoscopic lumbar foraminotomy. (**A**) Fluoroscopic view showing the initial working sheath placement (L4-5, left). The end of the working sheath should be opened to the outer foraminal window, while avoiding the ENR. (**B**) Endoscopic view showing the foraminal unroofing with the removal of SAP using endoscopic burrs. (**C**) Full-scale foraminal decompression was achieved by removing the foraminal ligaments and LF using endoscopic punches. (**D**) Final endoscopic view showing the released ENR. SAP, superior articular process; LF, ligamentum flavum; ENR, exiting nerve root.

**Figure 3 jcm-12-05745-f003:**
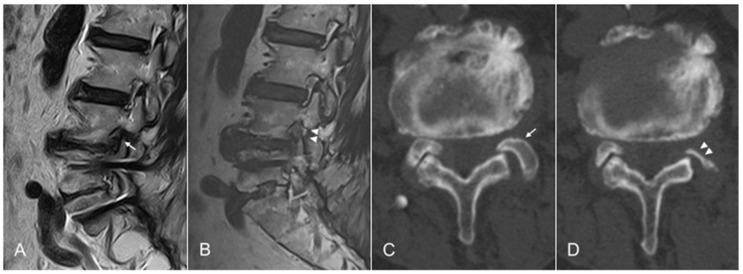
Representative case of an 81-year-old male patient with juxta-fusional foraminal stenosis. (**A**) Preoperative MRI showing foraminal stenosis at the L4-5 level (arrow) after fusion surgery at the L5-S1 level. (**B**) Postoperative MRI showing foraminal decompression following the removal of the bony and soft tissues compressing the exiting nerve root (arrowheads). (**C**) Preoperative CT images showing foraminal stenosis at the L4-5 level (arrow) after fusion surgery at the L5-S1 level. (**D**) Postoperative CT images at the critical level showing foraminal decompression following the removal of the bony and soft tissues compressing the exiting nerve root (arrowheads). However, most parts of the facet joint except the critical point are preserved. MRI, magnetic resonance image; CT, computed tomography.

**Figure 4 jcm-12-05745-f004:**
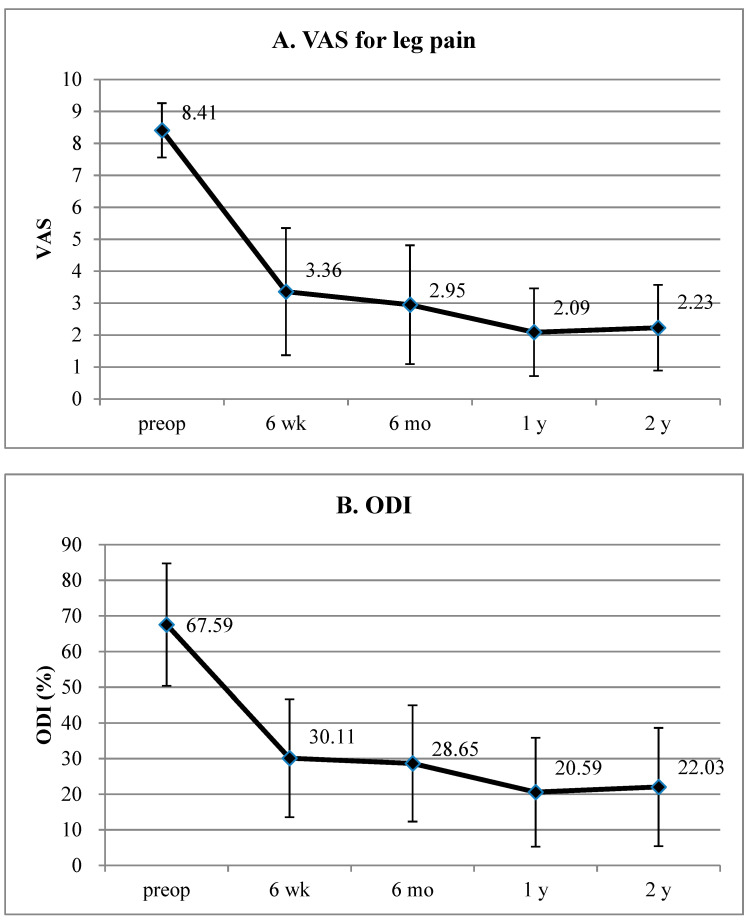
Clinical outcomes of TELF for juxta-fusional lumbar foraminal stenosis. (**A**) VAS pain score for radicular pain preoperatively and at 6 weeks, 6 months, 1 year, and 2 years postoperatively. (**B**) ODI scores preoperatively and at 6 weeks, 6 months, 1 year, and 2 years postoperatively. TELF, transforaminal endoscopic lumbar foraminotomy; VAS, visual analog scale; ODI, Oswestry disability index.

**Figure 5 jcm-12-05745-f005:**
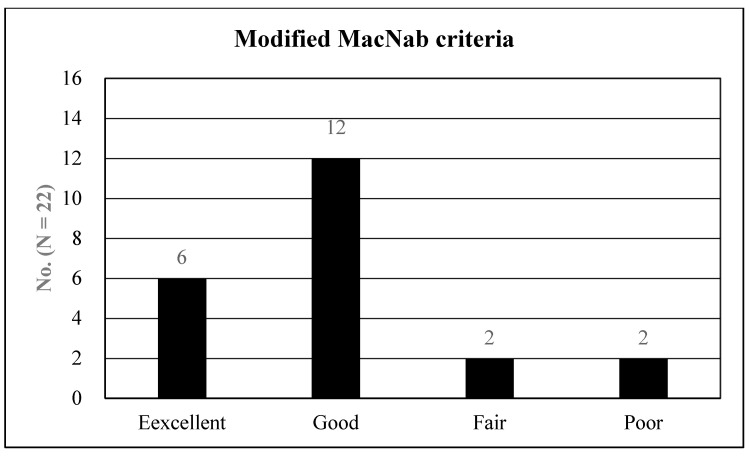
Global outcomes based on the modified MacNab criteria: the procedure outcomes were excellent in six patients (27.27%), good in 12 (54.55%), fair in two (9.09%), and poor in two (9.09%). Therefore, the rate of symptomatic improvement was 90.91%.

**Table 1 jcm-12-05745-t001:** Demographic characteristics.

Variables (*n* = 22)	No.	%
Sex		
Male	10	45.5
Female	12	54.5
Symptom-free period (primary fusion—radicular symptom)	50.32 ± 20.45 (mo)	
Symptom duration (radicular symptom—endoscopic procedure)	12.64 ± 7.77 (mo)	
Previous fusion level		
L3-4	3	13.6
L4-5	12	54.5
L5-S1	5	22.7
L3-4-5	1	4.5
L4-5-S1	1	4.5
Direction of adjacent disease		
Upper	13	59.1
Lower	9	40.9
Operative level		
L2-3	1	4.5
L3-4	7	31.8
L4-5	8	36.4
L5-S1	6	27.3

## Data Availability

The data presented in this study are available on request from the corresponding author.

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
