# Peer review of "Transforaminal Endoscopic Lumbar Foraminotomy for Juxta-Fusional Foraminal Stenosis"

_jcm, 2023, doi:10.3390/jcm12175745_

Round 1

Reviewer 1 Report

Lines 29-32: Suggest adding more references.

Lines 36-38: Please provide references.

Lines 39-42: Please provide references.

Line 205-6: Suggest including that impairment of cognitive function and Alzheimer's disease is very rare, or remove this part entirely.

Author Response

Author response

<Reviewer 1>

Lines 29-32: Suggest adding more references.

Lines 36-38: Please provide references.

Lines 39-42: Please provide references.

Line 205-6: Suggest including that impairment of cognitive function and Alzheimer's disease is very rare, or remove this part entirely.

Response: Thank you for your constructive review.

According to your recommendation, we added adequate references.

We also removed the part you pointed out. “Generally, the use of general anesthesia may increase the risk of adverse events, such as cardiovascular disorders, respiratory problems, blood loss, nausea, vomiting, nerve damage, impairment of cognitive function, and Alzheimer's disease.“

<Reviewer 2>

The authors of this manuscript describe their technique to treat foraminal stenosis that occurs in adjacent levels after lumber fusion surgery. The authors report on 22 patients treated by transforaminal endoscopic lumbar foraminotomy and generally report positive results. While this reviewer does not find any scientific flaws in their report, there are several points that need to be addressed.

Major points:

  1. The duration between primary surgery and onset of radiculopathy is not sufficiently described. This is a tremendously important point, because this directly influences the evaluation of whether the pathology can be designated as adjacent segment disease or not. Please describe the duration between primary surgery and onset of radiculopathy for all cases and include this information in Table 1.

Response: Thank you for your constructive advice. According to your recommendation, we added the following in the Results section and Table 1: “The mean duration between primary fusion surgery and the onset of radiculopathy was 50.32 months (range, 15 – 74 months). The mean symptom duration was 12.64 months (range, 3 – 25 months).” (Lines 168 – 170)

In Table 1.

Symptom-free period (primary fusion - radicular symptom)

50.32 ± 20.45 (mo)

Symptom duration (radicular symptom - endoscopic procedure)

12.64 ± 7.77 (mo)

  1. Adjacent segment disease following lumbar fusion surgery is reported to be higher than 10% if followed for a sufficiently long period. While the pathology seen in the adjacent segment is variable, it is usually central stenosis with or without spondylolisthesis. Invariably, foraminal stenosis is a minority of the pathology seen in adjacent segment disease. Therefore, the authors should describe the context of their subjects by disclosing the following: 1) number of lumbar fusion surgeries to begin with during the period their subjects received their primary surgery, along with 2) the number of cases diagnosed with all types of adjacent segment disease (and ideally the pathology, such as disc hernia, central stenosis, degenerative spondylolisthesis, and foraminal stenosis). 

Response: We agree with your opinion. However, evaluating the natural course and incidence of adjacent segment disease (ASD) was not the purpose of the study. Furthermore, we could not assess the natural course or incidence of ASD because most patients underwent fusion surgery at another institute, and they visited our clinic without detailed data.

We added the following in the 4.6. Limitations of the study - Discussion section.

“Finally, we could not evaluate the natural course or incidence of ASD after fusion surgery. Although foraminal stenosis after fusion surgery comprises a minority of the pathologies seen in ASD, we strictly included adjacent segment foraminal stenosis cases in our study. First, most of our patients underwent primary surgery at another institute. Only three of the 22 patients underwent fusion surgery at our hospital. Therefore, we could not obtain the clinical and radiological data at the time of the primary surgery in most patients. Second, the main purpose of this study was to evaluate the clinical outcome of the TELF technique for juxta-fusional foraminal stenosis cases. Central stenosis, intracanalicular disc herniation, or spondylolisthesis was not indicated for this procedure. Third, this study was retrospective, and most patients lacked detailed primary data. Therefore, we could not demonstrate the postoperative course or incidence of ASD after the primary surgery.” (Lines 277 – 286)

  1. In order to substantiate the authors’ claim of adjacent segment disease, the authors need to confirm that there were no cases with subclinical foraminal stenosis before the primary lumbar spinal fusion surgery. This reviewer highly suspects that a number of patients may have had similar foraminal stenosis on MRI images before surgery. The authors must directly confirm that no patients had significant foraminal stenosis before their primary surgery and state this within the text. 

Response: As mentioned before, this study was retrospective and we could not obtain the longitudinal data after the primary fusion surgery performed at other hospitals. However, all patients had some pain-free period after the primary fusion surgery and then began to experience radicular pains.

We described the pain-free period and symptom duration in the Results section. “The mean duration between primary fusion surgery and the onset of radiculopathy was 50.32 months (range, 15 – 74 months). The mean symptom duration was 12.64 months (range, 3 – 25 months).“ (Lines 168 – 170)

We also added the following in the Methods section. “The inclusion criteria were: 1) unbearable radicular leg pain despite more than three months of non-operative therapies; 2) at least 12 months of a pain-free period after the primary lumbar fusion surgery; 3) exiting nerve root (ENR) compression resulting from severe foraminal stenosis [14,15], at the level adjacent to the previous fusion surgery location as demonstrated by computed tomography (CT) and magnetic resonance imaging (MRI); 4) stable foraminal stenosis without significant segmental instability on dynamic lateral X-rays; and 5) foraminal stenosis, as the source of the radiculopathy, which was proven by thorough neurologic examination and selective nerve root block to the ENR. Patients with isolated low back pain, acute disc herniation, severe central stenosis, segmental instability, or other pathological conditions such as inflammation, infection, trauma, or tumors were excluded.“ (Lines 59 – 68)

Furthermore, pre-existing adjacent level foraminal stenosis does not affect the outcomes of lumber interbody fusion surgery [Shimamura Y 2023]

Shimamura Y, Kanayama M, Oha F, Tsujimoto T, Takana M, Hasegawa Y, Endo T, Hashimoto T. Pre-existing adjacent level foraminal stenosis does not affect the outcome of a single level lumbar interbody fusion. J Orthop Sci. 2023 Jul;28(4):719-723. doi: 10.1016/j.jos.2022.03.006. Epub 2022 Apr 22. PMID: 35469740.

We added the following in the Discussion section (4.3. TELF technique for juxta-fusional foraminal stenosis)

“Based on the pathogenesis, ASD can be categorized into three disease entities: ASD due to central stenosis, ASD due to foraminal stenosis, and ASD due to herniated disc [Shimamura Y 2023]. Theoretically, percutaneous endoscopic or key-hole decompression may be more suitable for foraminal stenosis than wide central stenosis [23]. Therefore, our endoscopic procedure focused on the foraminal pathologies at the juxta-fusional segment.” (Lines 221 – 225)

  1. While the authors state that the cases did not have any segmental instability, the CT images in Figure 3 show significant widening of the left facet joint undergoing decompression. First of all, this fact brings into question the authors’ definition of instability, because facet joint gaps usually denotes instability. Second, even if segmental instability was not apparent in dynamic radiographs, the fact remains that this fact gap should deter the authors from performing aggressive resection of the facet joint. In fact, although this reviewer acknowledges that the support of the facet joint may be intact, the post-decompression axial images of the remaining facet joint is worrisome. The authors need to explain their thoughts regarding this point. 

Response: Thank you for your essential point. The illustrated cut shows the undercutting of the superior articular process (SAP) and foraminal decompression. However, in the 3-dimensional aspect, the undercutting of the SAP was performed only at the critical level. Most parts of the facet joint could be preserved. That is one of the advantages of minimally invasive endoscopic decompression. Our series noted no clinical or radiological signs of further instability during the follow-up period. Unfortunately, we could not check postoperative 3-D CT in our series. Please consider this point. We added the following to the figure legends. “(d) Postoperative CT images at the critical level showing foraminal decompression following the removal of the bony and soft tissues compressing the exiting nerve root (arrowheads). However, most parts of the facet joint except the critical point are preserved.“ (Lines 150 – 153)

Minor points:

  1. The authors stress the ability to perform the TELF procedure percutaneously under local anesthesia as a potential benefit of the procedure, citing the risks of general anesthesia for elderly or high-risk patients. This point, while not dismissed, seems exaggerated considering that all of these patients underwent lumbar fusion surgery under general anesthesia. This reviewer suggests that the authors do not stress this point in this manuscript.

Response: We agree with your opinion. Open spine surgery under general anesthesia is still one of the gold-standard techniques. We removed this issue from the manuscript.

We revised the following in the Introduction section. “However, the invasiveness of this approach can lead to considerable morbidity or surgical trauma, especially in the elderly or patients with other medical illnesses.“ (Lines 36 – 37)

We also revised the following in the Discussion section (4.4. Pros and cons of TELF). “Second, the procedure can be performed under local anesthesia with intravenous sedation. Therefore, direct injury to the ENR can be avoided by monitoring patient feedback. Furthermore, elderly patients or patients at high risk of medical comorbidities can undergo TELF under general anesthesia without the risk of open surgery.“ (Lines 245 – 246)

  1. Many surgeons uncomfortable with foraminal stenosis elect to perform PLIF/TLIF/LLIF procedures to treat primary foraminal stenosis. This reviewer suspects that the authors perform their TELF procedure successfully in primary foraminal stenosis patients, but am puzzled that the authors do not mention or reference their previous reports on primary foraminal stenosis. I suggest the authors address this point. 

Response: Thank you for your advice. We have published articles on the technique and clinical outcomes of TELF. The number of our articles cited in the manuscript should be limited because the journal office limited the self-citation rate to less than 10%. Please consider this.

We added the following in the Introduction section. “We have also demonstrated our techniques and clinical results of TELF for primary lumbar foraminal stenosis [20,23,24].” (Lines 42 – 43)

To facilitate the reviewer’s understanding, we have listed our references on the TELF technique.

Ahn Y, Lee SH, Park WM, Lee HY. Posterolateral percutaneous endoscopic lumbar foraminotomy for L5-S1 foraminal or lateral exit zone stenosis. Technical note. J Neurosurg. 2003 Oct;99(3 Suppl):320-3. doi: 10.3171/spi.2003.99.3.0320. PMID: 14563153.

Ahn Y, Kim WK, Son S, Lee SG, Jeong YM, Im T. Radiographic Assessment on Magnetic Resonance Imaging after Percutaneous Endoscopic Lumbar Foraminotomy. Neurol Med Chir (Tokyo). 2017 Dec 15;57(12):649-657. doi: 10.2176/nmc.oa.2016-0249. Epub 2017 Oct 19. PMID: 29046504; PMCID: PMC5735228.

Ahn Y, Keum HJ, Son S. Percutaneous Endoscopic Lumbar Foraminotomy for Foraminal Stenosis with Postlaminectomy Syndrome in Geriatric Patients. World Neurosurg. 2019 Oct;130:e1070-e1076. doi: 10.1016/j.wneu.2019.07.087. Epub 2019 Jul 16. PMID: 31323406.

Ahn Y, Keum HJ, Shin SH, Choi JJ. Laser-assisted endoscopic lumbar foraminotomy for failed back surgery syndrome in elderly patients. Lasers Med Sci. 2020 Feb;35(1):121-129. doi: 10.1007/s10103-019-02803-7. Epub 2019 May 17. PMID: 31102002.

Ahn Y, Oh HK, Kim H, Lee SH, Lee HN. Percutaneous endoscopic lumbar foraminotomy: an advanced surgical technique and clinical outcomes. Neurosurgery. 2014 Aug;75(2):124-33; discussion 132-3. doi: 10.1227/NEU.0000000000000361. PMID: 24691470; PMCID: PMC4086756.

Rhee DY, Ahn Y. Full-Endoscopic Lumbar Foraminotomy for Foraminal Stenosis in Spondylolisthesis: Two-Year Follow-Up Results. Diagnostics (Basel). 2022 Dec 13;12(12):3152. doi: 10.3390/diagnostics12123152. PMID: 36553159; PMCID: PMC9777364.

Ahn Y, Park HB, Yoo BR, Jeong TS. Endoscopic lumbar foraminotomy for foraminal stenosis in stable spondylolisthesis. Front Surg. 2022 Nov 10;9:1042184. doi: 10.3389/fsurg.2022.1042184. PMID: 36439521; PMCID: PMC9687795.

Ahn Y, Lee SG. Percutaneous endoscopic lumbar foraminotomy: how I do it. Acta Neurochir (Wien). 2022 Mar;164(3):933-936. doi: 10.1007/s00701-022-05114-z. Epub 2022 Jan 12. PMID: 35020086.

<Additional correction>

According to the editor’s recommendation, we further made the following revisions.

  1. We revised the surgical procedures in more detail.

             2.2.1. Fluoroscopy-guided transforaminal approach

“The location of the nerve root in the neural foramen and the severity of neural compression should be checked in the imaging studies. The initial trajectory should be directed to the foraminal pathologies while avoiding direct injury to the nerve root.” (Lines 90 – 93)

“Ideally, the working sheath should be firmly engaged in the foramen while avoiding the ENR to obtain a sufficient surgical field for foraminal decompression. Then, the working channel endoscope can be introduced to view the foraminal anatomies, including ENR with some perineural fat, disc surface, hypertrophic foraminal ligaments, and facet joints.“ (Lines 98 – 102)

             2.2.2. Bony unroofing under endoscopic visualization

“Proper exposure of the facet joint is essential for effective bony unroofing. Both SAP and the pedicle, including the synovium, should be exposed. Then, the SAP resection can proceed along the synovium surface. Finally, the surgeon can reach the axillary epidural space and LF.“ (Lines 108 – 111)

  1. We added a brief graphical abstract.
  2. We added the following in the Discussion section (4.7. Future perspective)

“The working channel endoscope with a bigger working space enables more potent instruments. Steerable or navigable forceps and punches can reach the corner-side pathologies. Various articulating burrs will make the bony resection safer and more efficient. A high-definition monitor system will make the surgeon perform the procedure more precisely. The spinal navigation system will be applied to the endoscopic spine surgery to increase the accuracy of the surgery and to decrease the radiation exposure during the surgery.” (Lines 292 – 298)

Reviewer 2 Report

The authors of this manuscript describe their technique to treat foraminal stenosis that occurs in adjacent levels after lumber fusion surgery. The authors report on 22 patients treated by transforaminal endoscopic lumbar foraminotomy and generally report positive results. While this reviewer does not find any scientific flaws in their report, there are several points that need to be addressed.

Major points:

1.     The duration between primary surgery and onset of radiculopathy is not sufficiently described. This is a tremendously important point, because this directly influences the evaluation of whether the pathology can be designated as adjacent segment disease or not. Please describe the duration between primary surgery and onset of radiculopathy for all cases and include this information in Table 1.

2.     Adjacent segment disease following lumbar fusion surgery is reported to be higher than 10% if followed for a sufficiently long period. While the pathology seen in the adjacent segment is variable, it is usually central stenosis with or without spondylolisthesis. Invariably, foraminal stenosis is a minority of the pathology seen in adjacent segment disease. Therefore, the authors should describe the context of their subjects by disclosing the following: 1) number of lumbar fusion surgeries to begin with during the period their subjects received their primary surgery, along with 2) the number of cases diagnosed with all types of adjacent segment disease (and ideally the pathology, such as disc hernia, central stenosis, degenerative spondylolisthesis, and foraminal stenosis). 

3.     In order to substantiate the authors’ claim of adjacent segment disease, the authors need to confirm that there were no cases with subclinical foraminal stenosis before the primary lumbar spinal fusion surgery. This reviewer highly suspects that a number of patients may have had similar foraminal stenosis on MRI images before surgery. The authors must directly confirm that no patients had significant foraminal stenosis before their primary surgery and state this within the text. 

4.     While the authors state that the cases did not have any segmental instability, the CT images in Figure 3 show significant widening of the left facet joint undergoing decompression. First of all, this fact brings into question the authors’ definition of instability, because facet joint gaps usually denotes instability. Second, even if segmental instability was not apparent in dynamic radiographs, the fact remains that this fact gap should deter the authors from performing aggressive resection of the facet joint. In fact, although this reviewer acknowledges that the support of the facet joint may be intact, the post-decompression axial images of the remaining facet joint is worrisome. The authors need to explain their thoughts regarding this point. 

Minor points:

1.     The authors stress the ability to perform the TELF procedure percutaneously under local anesthesia as a potential benefit of the procedure, citing the risks of general anesthesia for elderly or high-risk patients. This point, while not dismissed, seems exaggerated considering that all of these patients underwent lumbar fusion surgery under general anesthesia. This reviewer suggests that the authors do not stress this point in this manuscript.

2.     Many surgeons uncomfortable with foraminal stenosis elect to perform PLIF/TLIF/LLIF procedures to treat primary foraminal stenosis. This reviewer suspects that the authors perform their TELF procedure successfully in primary foraminal stenosis patients, but am puzzled that the authors do not mention or reference their previous reports on primary foraminal stenosis. I suggest the authors address this point. 

There are no major English problems to point out.

Author Response

(The authors gave the same response as above.)

Round 2

Reviewer 2 Report

The authors have addressed all concerns of this reviewer. The additional information has improved the understanding of the included cases.

This reviewer can now recommend publication of this manuscript in JCM.